# Self-assembled three dimensional network designs for soft electronics

Kyung-In Jang[1,2,*], Kan Li[3,*], Ha Uk Chung[1,4,*], Sheng Xu[5], Han Na Jung[1], Yiyuan Yang[1], Jean Won Kwak[1], Han Hee Jung[2], Juwon Song[2], Ce Yang[6], Ao Wang[3,6], Zhuangjian Liu[7], Jong Yoon Lee[1,2], Bong Hoon Kim[1], Jae-Hwan Kim[1], Jungyup Lee[1], Yongjoon Yu[1], Bum Jun Kim[1], Hokyung Jang[1], Ki Jun Yu[8], Jeonghyun Kim[9], Jung Woo Lee[10], Jae-Woong Jeong[11], Young Min Song[12], Yonggang Huang[3], Yihui Zhang[6] & John A. Rogers[13]

Low modulus, compliant systems of sensors, circuits and radios designed to intimately interface with the soft tissues of the human body are of growing interest, due to their emerging applications in continuous, clinical-quality health monitors and advanced, bioelectronic therapeutics. Although recent research establishes various materials and mechanics concepts for such technologies, all existing approaches involve simple, two-dimensional (2D) layouts in the constituent micro-components and interconnects. Here we introduce concepts in three-dimensional (3D) architectures that bypass important engineering constraints and performance limitations set by traditional, 2D designs. Specifically, open-mesh, 3D interconnect networks of helical microcoils formed by deterministic compressive buckling establish the basis for systems that can offer exceptional low modulus, elastic mechanics, in compact geometries, with active components and sophisticated levels of functionality. Coupled mechanical and electrical design approaches enable layout optimization, assembly processes and encapsulation schemes to yield 3D configurations that satisfy requirements in demanding, complex systems, such as wireless, skin-compatible electronic sensors.

[1] Frederick Seitz Materials Research Laboratory, Department of Materials Science and Engineering, University of Illinois at Urbana–Champaign, Urbana, Illinois 61801, USA. [2] Department of Robotics Engineering, Daegu Gyeongbuk Institute of Science and Technology (DGIST), Daegu 42988, South Korea. [3] Departments of Civil and Environmental Engineering, Mechanical Engineering, and Materials Science and Engineering, Northwestern University, Evanston, Illinois 60208, USA. [4] Department of Electrical Engineering and Computer Science, Northwestern University, Evanston, Illinois 60208, USA. [5] Department of NanoEngineering, University of California at San Diego, La Jolla, California 92093, USA. [6] Department of Engineering Mechanics, Center for Mechanics and Materials, AML, Tsinghua University, Beijing 100084, China. [7] Institute of High Performance Computing, Singapore 138632, Singapore. [8] School of Electrical and Electronic Engineering, Yonsei University, Seoul 03722, Republic of Korea. [9] Department of Electronics Convergence Engineering, Kwangwoon University, Seoul 01897, Republic of Korea. [10] Department of Materials Science and Engineering, Pusan National University, Busan 46241, Republic of Korea. [11] Department of Electrical, Computer and Energy Engineering, University of Colorado, Boulder, Colorado 80309, USA. [12] School of Electrical Engineering and Computer Science, Gwangju Institute of Science and Technology, Gwangju 61005, Republic of Korea. [13] Departments of Materials Science and Engineering, Biomedical Engineering, Chemistry, Mechanical Engineering, Electrical Engineering and Computer Science, Neurological Surgery, Center for Bio-Integrated Electronics, Simpson Querrey Institute for BioNanotechnology, McCormick School of Engineering and Feinberg School of Medicine, Northwestern University, Evanston, Illinois 60208, USA. * These authors contributed equally to this work. Correspondence and requests for materials should be addressed to Y.H. (email: y-huang@northwestern.edu) or to Y.Z. (email: yihuizhang@tsinghua.edu.cn) or to J.A.R. (email: jrogers@northwestern.edu).

Rapid advances in the development of precision chem/biosensors, low-power radio communication systems, efficient energy harvesting/storage devices, high-capacity memory technologies and miniaturized electronic/optoelectronic components create opportunities for qualitatively expanding the ways that microsystem technologies can be integrated with the human body for treating disease states and monitoring health status[1–5]. Realizing this potential requires not only advances in the components but also in the strategies for their collective integration into systems that offer stable, long-term operation at intimate biotic/abiotic interfaces[1,3,4]. Promising research in this direction focuses on development of materials and mechanics concepts to enable system-level properties that are mechanically and geometrically matched to those of the soft tissues of the human body[6–25]. Although previously reported approaches provide significant utility in this context[12,14,26–42], they all rely on planar, two-dimensional (2D) layouts of the functional elements and electrical interconnects, where the 2D geometries and/or materials define the essential physical characteristics. The work reported here pursues a different strategy, in which system architectures adopt engineered, three-dimensional (3D) designs to provide properties that circumvent intrinsic limitations associated with traditional, 2D counterparts. Specifically, combined experimental and theoretical results demonstrate that open networks of 3D microscale helical interconnects offer nearly ideal mechanics for soft electronic systems that embed chip-scale components, by virtue of model, spring-like behaviours similar to those in man-made[43–45] (for example, coil spring) and biological[46–50] (for example, tendrils) analogues. Furthermore, scalable methods for forming the required 3D mesostructures together with systematic approaches for defining mechanically and electrically optimized layouts allow immediate application to complex systems. Wireless sensor platforms capable of physiological status monitoring in soft, skin-mounted formats provide demonstration examples. These findings could find broad utility in many classes of soft microsystems technologies.

## Results

### Mechanics of 3D helical coils.
The assembly approach builds on recently introduced concepts in deterministically controlled buckling processes[51,52], in which initial 2D structures spontaneously transform into desired 3D shapes. Figure 1a presents finite element analyses (FEA, see Methods section for details) for the formation of an extended, spiral network of 3D helical microstructures from a corresponding 2D precursor that takes the form of a collection of filamentary serpentine ribbons (widths: 50 μm; multilayer construction: 2.5 μm PI/1.0 μm Au/5 nm Cr/2.5 μm PI). Each unit cell consists of two identical arcs with central angle $\theta_0$ and radius $r_0$. The two ends include small discs that form strong covalent siloxane bonds to an underlying elastomeric silicone substrate in a state of biaxial prestrain; other regions adhere only through weak van der Waals interactions. Compressive forces induced by releasing the prestrain cause the 2D precursor to geometrically transform, through a coordinated collection of in-plane and out-of-plane translational and rotational motions, into an engineered 3D configuration via controlled buckling deformations[51,52]. Specifically, each unit cell transforms into a single turn of a corresponding 3D helical microcoil whose pitch ($p$) depends on the serpentine geometry and the magnitude of the prestrain, $\varepsilon_{pre}$, according to $p = 4r_0 \sin(\theta_0/2)/(1 + \varepsilon_{pre})$. The overall shape of coil follows mainly from $\varepsilon_{pre}$ and $\theta_0$ (Supplementary Fig. 1). Optical images of a representative structure appear in Fig. 1b. The key dimensions match those from FEA. For example, the experimental and FEA results for the pitch, height of the microcoils are $1,150 \pm 40$ μm, $530 \pm 50$ μm and 1,170, 520 μm, respectively.

Although previous studies establish the utility of planar serpentine interconnects in soft electronics, the existence of sharp, localized stress concentrations that follow from their 2D formats and their physical coupling to the substrate limit performance for systems that require low modulus, elastic mechanics in compact designs. By comparison, 3D helical microstructures avoid this unfavourable mechanics due to smoothly varying, uniform distributions of deformation-induced stresses that follow directly from their 3D layouts. The result enables exceptionally high levels of stretchability and mechanical robustness, without the propensity for localized crack formation or fracture, as supported by results in Fig. 1c,d and Supplementary Fig. 2. For purposes of quantitative comparison, consider a 2D serpentine interconnect with the same material composition (PI/Au/PI) and key geometric parameters (Supplementary Fig. 2a), including the width ($w$), the thickness ($t_{metal}$ and $t_{PI}$), the span ($S$) and the amplitude ($A$), as a corresponding 3D helix. Here the number of unit cells in the 2D serpentine is selected such that its total trace length ($l_{total}$) is approximately the same as that of the 3D counterpart (10.68 mm, $\theta_0 = 180^\circ$ and $r_0 = 425$ μm). The limit of elastic stretchability, defined as the maximum overall dimensional change below which structural deformations can recover completely, corresponds to the point at which the constituent materials undergo plastic deformation at the locations of highest Mises stress. Results for analogous 2D and 3D systems appear in Supplementary Fig. 2b. Due to the absence of stress concentrations, the elastic stretchability of the 3D helices significantly exceeds that of the 2D serpentines. The enhancement corresponds to a factor of ∼3 for $\varepsilon_{pre} = 50\%$, and this factor increases continuously with $\varepsilon_{pre}$ until it reaches ∼9.5 for $\varepsilon_{pre} = 300\%$ (Supplementary Fig. 2b).

These improvements follow from qualitatively distinct deformation mechanisms in 3D compared to 2D layouts. Figure 1c illustrates the nearly ideal, spring-like mechanics that characterize responses in 3D helices. Here, deformations are almost completely decoupled from those of elastomeric substrate and the cross-sectional maximum Mises stress ($\sigma_{Mises}^{cross-section}$) are spatially uniform, except for small regions near the bonding sites. Quantitative results from FEA appear in Fig. 1e, which shows the maximum Mises stress for each cross section along the natural coordinate normalized by the arc length. Deformations of the 2D serpentine lead to sharp, unavoidable stress concentrations at the arc regions, as illustrated by the results in Fig. 1f. The ratio of the peak stress $[\text{Peak}(\sigma_{Mises}^{cross-section})]$ to its mean value $[\text{Mean}(\sigma_{Mises}^{cross-section})]$ serves as a metric for the magnitude of this concentration. This ratio is only ∼1.2 for the 3D helices under both applied strains of 25 and 50%, which is nearly an order of magnitude smaller than the ratio (∼9.8) for the 2D serpentines. The non-uniformity in the stress distribution can be characterized by the average absolute deviation relative to the mean value, as given by $F_{non-uniformity} = \frac{\int_0^1 |\sigma_{Mises}^{cross-section}(\bar{S}) - \text{Mean}(\sigma_{Mises}^{cross-section})|d\bar{S}}{\text{Mean}(\sigma_{Mises}^{cross-section})}$, where $\bar{S}$ denotes the natural arc coordinate normalized by the total trace length of entire interconnect. This dimensionless factor also highlights the qualitative differences (for example, 0.15 versus 1.06 for 50% stretch) between behaviours of the 3D helical and 2D serpentine microstructures. Even when compared to fractal 2D designs, in which microfluidic enclosures afford mechanical decoupling from the substrate[14] or to bar-type 2D serpentines, in which planar, scissor-like mechanics dominates[53], the 3D helical geometry is superior due to the spring-like responses and associated uniformity in the stress distribution (see Supplementary Figs 3–7 for details). In particular, 3D helical interconnects outperform 2D serpentine designs that undergo buckling deformations in the form of local wrinkling, typically by a factor >2.3 for representative prestrains (>100%). As compared to thick serpentine interconnects that are

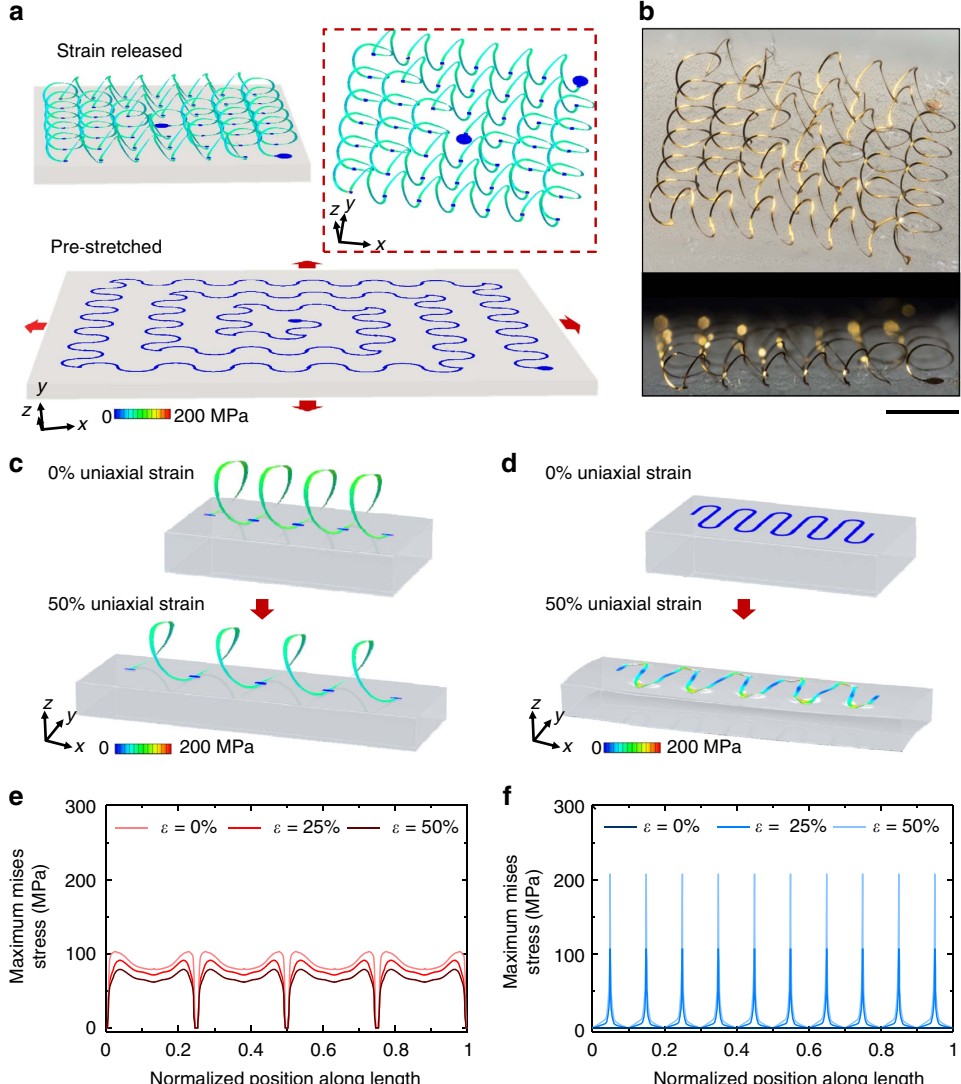

**Figure 1 | Assembly of conductive 3D helical coils and their mechanical properties.** (**a**) Process for assembly illustrated by finite element analysis (FEA). A 2D filamentary serpentine structure bonded at selected locations to an underling, bi-axially stretched ($\varepsilon_{pre}$) soft elastomeric substrate (pre-streched). Corresponding 3D helical coils formed by relaxation of the substrate to its initial, unstretched state (strain released). The colour represents the magnitude of Mises stress in the metal layer. (**b**) Angled and cross-sectional optical images of an experimentally realized structure. The traces consist of lithographically defined multilayer ribbons of polyimide/Au or Cu/polyimide bonded to a silicone substrate. Scale bar, 1 mm. FEA results for the deformations and distributions of Mises stress in a 3D coil (**c**) and a 2D serpentine (**d**) with similar geometries at 0 and 50% uniaxial strain. (**e**) Distribution of maximum Mises stress for each cross section along the natural coordinate normalized by the arc length, for the 3D helical coil in **c**, for three different levels of applied strain (0, 25, 50%). (**f**) Similar results for the case of the 2D serpentine in **d**.

dominated by planar, scissor-like mechanics, 3D helical interconnects with a sufficient prestrain (for example, >100%) exhibit a considerable enhancement (for example, >1.8 times) in the elastic stretchability. Similar improvements, at an even greater factor, apply relative to 2D interconnects with fractal-inspired layouts (Supplementary Fig. 7).

**Design of 3D interconnect network of helical coils.** The versatility in layouts that can be realized by the assembly approach outlined in Fig. 1, the scalability of this process to large areas, and the excellent mechanical behaviours of the 3D helical designs facilitate straightforward implementation even in complex systems (see Supplementary Figs 8–15). The optical image of Fig. 2a presents an example that consists of ~50 separate chip-scale electrical components, ~250 distinct 3D

helical interconnects (PI 2.5 μm/Au 1 μm/Cr 5 nm/PI 2.5 μm multilayers with 50 μm in width for each line; PI 2.5 μm/Au 1 μm/Cr 5 nm/PI 2.5 μm/Au 1 μm/Cr 5 nm/PI 2.5 μm multilayers with 50 μm in width for each crossing point) adhered at ~500 bonding sites to an elastomeric substrate ($E = 20$ kPa; Ecoflex 00–50, Smooth-on, USA; ~4 cm diameter at rest), all encapsulated with an ultra-low-modulus elastomer ($E = 3$ kPa; Silbione 4717A/B, Bluestar Silicones, France). As described subsequently and demonstrated in Supplementary Movie 1, this platform provides wireless, battery-free capabilities in continuous monitoring of physiological health from mounting locations on the skin. The inset shows the device in a complex state of deformation to illustrate the soft, skin-like physical properties. The overall layout employs a spider-web-like geometry (Supplementary Figs 16–18), selected to avoid failure at any point in the system, to ensure uniform and extreme levels of

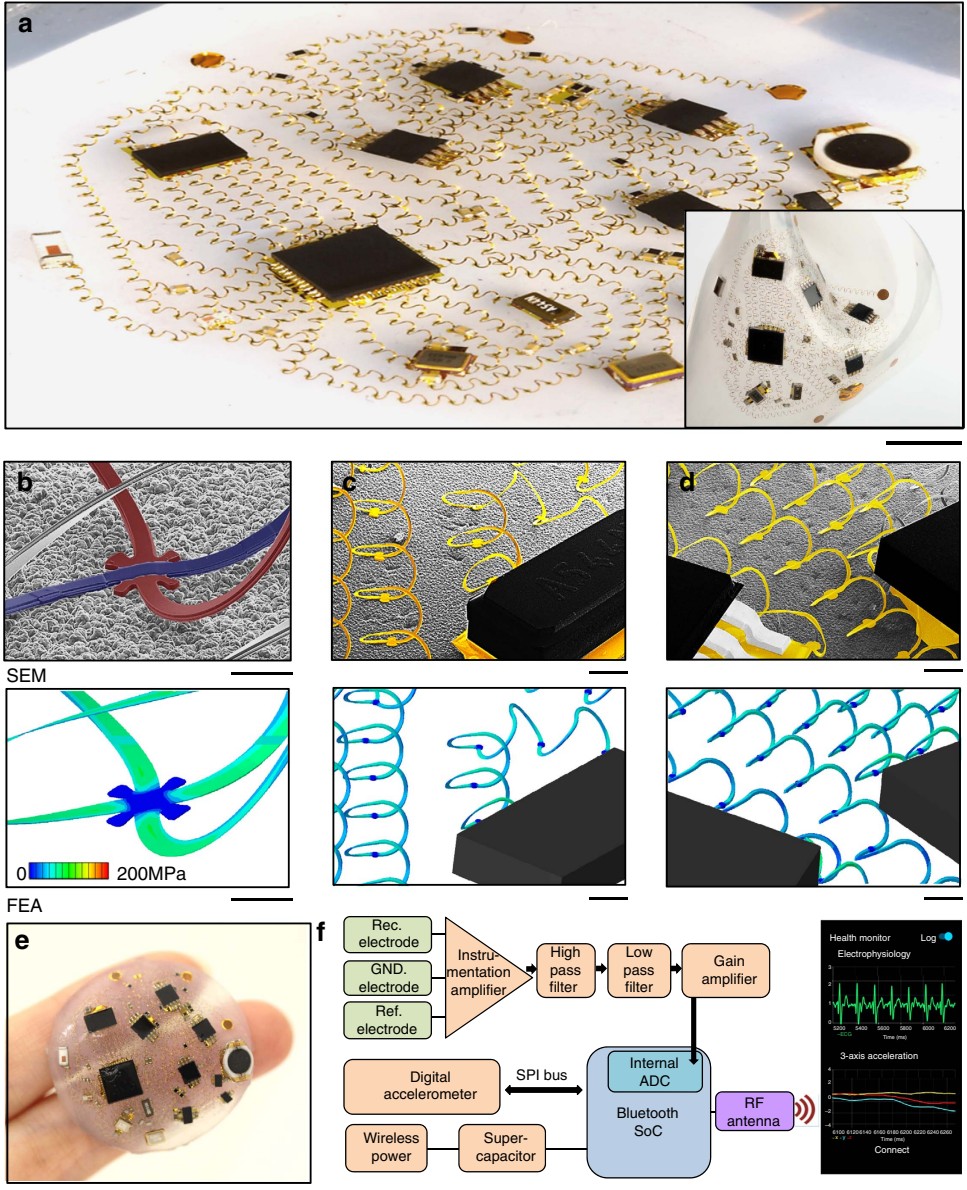

**Figure 2 | Three-dimensional network of helical coils as electrical interconnects for soft electronics.** (**a**) Optical image of the system at a bi-axially stretched state of 50%, showing ~250 3D helices, ~500 bonding sites, ~50 component chips and elastomers for full encapsulation (Encapsulation material: Silbione 4717A/B, Bluestar Silicones, France; $E = 3$ kPa; bottom substrate: Ecoflex 00-50A/B, Smooth-on, USA; $E = 20$ kPa) Inset: optical image of the device under a complex state of deformation. Scale bar, 5 mm. Scanning electron micrographs and corresponding FEA results of representative regions of the 3D network, including (**b**) electrically isolated crossing points and (**c**,**d**) interfaces with chip components. Scale bars, 100 μm (**b**), 1 mm (**c**) and 1 mm (**d**). The widths and thicknesses of all coils throughout this system are 50 and 1 μm, respectively. (**e**) Optical image of a device supported by two fingers. (**f**) Block diagram of the functional components for a set of electrophysiological (EP) sensors with an analogue signal processing unit that includes filters and amplifiers, a three-axial digital accelerometer, a Bluetooth system on a chip (SoC) for signal acquisition and wireless communication, and a wireless power transfer system for battery-free operation. The signal acquisition involves sampling of the EP sensor output through an internal analogue-to-digital converter (ADC) and data acquisition of the accelerometer output via a serial peripheral interface (SPI). When operated using a custom graphical user interface on a smart phone, this system can capture and transmit a range of information related to physiological health, as shown in Supplementary Movie 1.

stretchability and bendability in any direction and to minimize the overall system size. This unusual layout follows from a rigorous, systematic design approach that leverages FEA modelling. Specifically, for any given design, FEA allows rapid identification of locations of high Mises stresses and physical collisions between the different regions of the interconnect network and/or individual components, under various states of deformation within a desired range of strains. Under constraints of interconnect length and connectivity set by considerations in

electrical performance (details described below), an iterative process guided by FEA allows optimization of all relevant parameters, including the magnitude of the prestrain in the assembly process, the spatial layouts of the components, the configurations of the bonding sites and the geometries of the 2D serpentine precursors. The use of serpentine unit cells with fixed dimensions across the entire circuit simplifies the design process and yields uniform distributions of Mises stresses. Overlaid on these mechanical considerations is a set of electrical requirements.

For example, the length of the line that supplies power must be minimized to reduce resistive dissipation and electromagnetic noise associated with radio frequency power delivery. A large decoupling capacitor (22 μF) at the power source and smaller bypass capacitors at each branch of the supply route, all of which

have low equivalent series resistance, help to suppress this noise. The antenna and the electrodes for sensing electrophysiological signals must be spatially separated from the power line to minimize electromagnetic interference, but they must also enable impedance matching (50 Ω). These and other coupled

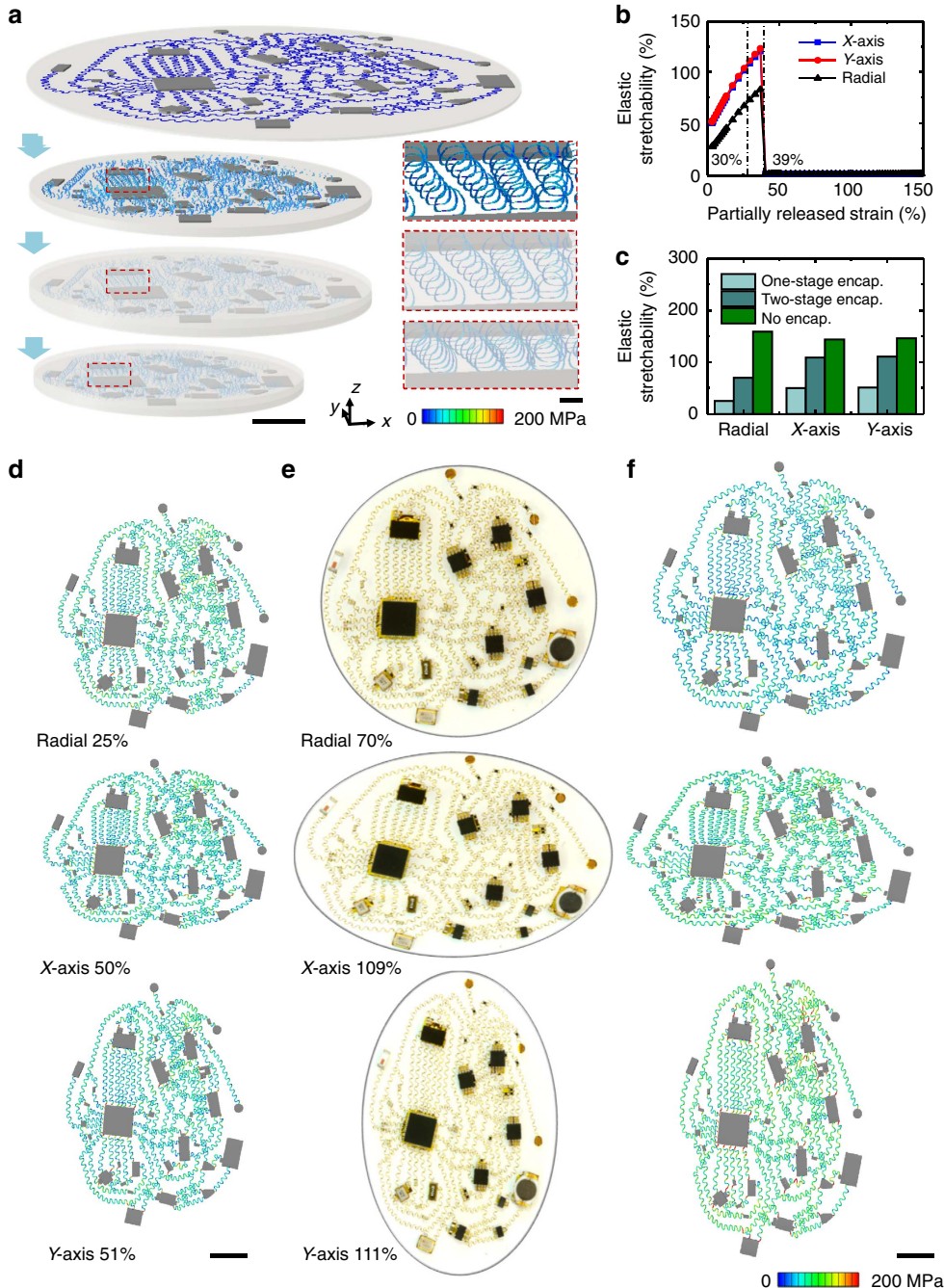

**Figure 3 | Strategy for soft encapsulation and system-level mechanics.** (**a**) Process for soft encapsulation illustrated with FEA results: first, a collection of electronic components joined by an interconnect network in a 2D serpentine design bond at selective sites to a bi-axially prestrained soft elastomeric substrate (Ecoflex 00-50A/B, Smooth-on, USA; $E = 20$ kPa). Partial release of the prestrain mechanically transforms the 2D serpentines into 3D helices. Coating the entire structure with a thin, low-modulus silicone elastomer (Silbione 4717A/B, Bluestar Silicones, France; $E = 3$ kPa) defines the encapsulation layer. Finally, releasing the remaining prestrain enhances the 3D geometry of the helices and completes the process. The insets correspond to magnified views of a local region. (**b**) Elastic stretchability of a 3D circuit system encapsulated at different states of partially released strain in this two-stage encapsulating process (blue for uniaxial stretching along $X$ axis; red for uniaxial stretching along $Y$ axis; black for radial stretching). (**c**) Bar graph of the elastic stretchability for a 3D circuit system formed by using a one-stage encapsulating process, two-stage encapsulating process and no encapsulation. (**d**) System-level deformation and distribution of stresses determined by FEA with encapsulation introduced after (**d**) full and (**f**) partial release of the prestrain. In all of the FEA images, the colour represents the magnitude of Mises stress in the metal layer. (**e**) Optical images of a device deformed in similar ways. Scale bars, 1 cm (**a**–**d**). The scale bar of the inset of (**a**) is 1 mm.

considerations in mechanics and electronics underpin the optimized design shown in Fig. 2a.

The scanning electron microscope (SEM) images in Fig. 2b–d highlight some key representative regions of this system, including heterogeneously integrated structures of 3D microcoils, chip-scale components and the supporting elastomeric substrate. In terms of the mechanics, these frames highlight the level of agreement between experiment and FEA predictions across the entire structure. Figure 2b illustrates electrical crossovers enabled by the multilayer construction of the traces. As shown in Fig. 2b–d, the bonding sites retain an undeformed shape due to the low modulus of the underlying elastomer and the relatively large thickness ($\sim 6\,\mu$m) of the interconnect structure. The result leads to low levels of Mises stresses at these locations. The chip-scale components prevent corresponding near-surface regions of the elastomer from relaxing after release of the prestrain, thereby inducing a certain level of strain concentration. As such, the portions of the helical interconnects that join directly to the chips undergo a slightly enhanced compression, consistent with the increased Mises stresses in Fig. 2d. The FEA results in the lower images show the stress field associated with the 3D microcoil. The maximum computed stress in the metal layer across the entire 3D interconnect structure is $\sim 130\,$MPa. This value corresponds to $\sim 0.19\%$ strain, which is below the yielding point ($\sim 0.3\%$).

Delamination has the potential to occur at the chip/substrate interface during the process of mechanical-guided assembly, as shown in experimental results (Supplementary Fig. 19). Theoretically, delamination occurs more readily as the prestrain increases or the effective rigidity of the chip increases. Such trends are in qualitative agreement with experimental observations (Supplementary Fig. 19), in which the delamination area increases as the effective rigidity of the chip increases. In addition, delamination occurs mainly at the periphery regions of the chips. This type of partial delamination does not affect the bonding conditions between electrical interconnects and substrate, and thereby has a minor effect on the integrity of the entire device.

As depicted in Fig. 2e,f, the device functions as a soft, skin-mounted technology with important capabilities in monitoring of well-known parameters relevant to physiological health status. Wireless delivery of power to a receiver antenna charges a supercapacitor which then supplies regulated power to the entire system. Sensors allow quantitative measurements of motion, via accelerometry, and electrophysiology, via skin-interfaced electrodes. Raw data pass through a collection of analogue filters and amplifiers prior to wireless transmission using a Bluetooth protocol to a customized app running on a smart phone. Details appear in the Methods and Supplementary Figs 20–23.

**Soft encapsulation strategy for enhanced mechanics.** Practical applications demand encapsulating layers to protect the active and passive components and the interconnects. As compared with liquid encapsulation[14], the solid encapsulation approach adopted in this paper avoids the potential for leakage or evaporation. As in previously reported systems that use 2D serpentines, ultra-low-modulus elastomers are attractive due to the minimal mechanical constraints that they impose on the motions of the components and interconnect networks. Typically, introduction of this encapsulation material occurs as a final step in the fabrication. An alternative, improved strategy that naturally follows from the 3D assembly process outlined in Fig. 1 involves applying this material as a liquid precursor at a state of partial release of the substrate prestrain, $\varepsilon_{encap}$, crosslinking it into a solid form and then completing the release, as shown in Fig. 3a. This last step deforms the cured material in a manner that softly embeds the

3D helical interconnects via mechanical interactions during final release. As illustrated by experimental results (Supplementary Fig. 24), the 3D buckled structures in the system maintain their original forms during the encapsulation process because their equivalent moduli are sufficiently high that they are not deformed by shear forces associated with the viscosity of the uncured elastomer or liquid.

Quantitative mechanics modelling reveals significant associated improvements in the elastic stretchability (Fig. 3b,c), compared to the usual case of encapsulation at the final stage of fabrication and assembly. Specifically, the enhancement corresponds to a factor of $\sim 2.2$ and 2.8 times for uniaxial and radial stretching, respectively, when $\varepsilon_{encap} = 30\%$ and $\varepsilon_{pre} = 150\%$ for the device in Fig. 2a ($E_{substrate} = 20\,$kPa and $E_{encapsulation} = 3\,$kPa). Modelling is critical for the practical implementation of this concept, simply because the use of $\varepsilon_{encap}$ above a certain threshold ($> 39\%$ for the case examined here) can lead to plastic yielding of the interconnects before full release (Fig. 3b). Near this limit, the elastic stretchability of the encapsulated 3D circuit system approaches that of the unencapsulated counterpart, particularly for uniaxial stretching (for example, 120 versus 144% for $X$ axis and 123% versus 146% for $Y$ axis) (see Supplementary Discussion for more discussions). By comparison, encapsulation at $\varepsilon_{encap} = 0$ (that is, encapsulation after complete assembly) retains only limited levels of stretchability (for example, $\sim 50\%$ for $X$ axis, 51% for $Y$ axis, and 25% for radial stretching, respectively), as shown in Fig. 3d. The deformation mode in Fig. 3e is, in fact, close to that of an ideal, unencapsulated system (Supplementary Fig. 25). In this construction, a radial strain of 70% induces maximum Mises stress of only $\sim 199\,$MPa (corresponding to its yield strain, $\sim 0.3\%$) in the active materials (Au of the interconnects), consistent with reversible, elastic behaviours. Furthermore, the stress distribution is uniform across most of the circuit (Fig. 3e), as additional evidence of the efficient mechanics.

The modulus of the top encapsulation has a significant effect on the elastic stretchability of the 3D helical interconnects (Supplementary Fig. 26). A low-modulus elastomer (Silbione) reduces mechanical constraints on the deformations of helical interconnects, thereby achieving a much higher stretchability than possible with elastomers with higher modulus values (for example, Ecoflex). The circuit can be designed with improved density via the use of increased prestrain during the 3D assembly process. By increasing the prestrain from 150 to 200%, the area of entire circuit can be reduced to $\sim 70\%$ of the original area. The elastic stretchability undergoes a relatively small corresponding reduction, as in Supplementary Fig. 27.

The devices might encounter compression along the thickness direction during practical use, associated with physical contact or normal impact. Quantitative modelling of the encapsulated helical interconnects provides insights into the mechanics associated with such situations. For a variety of prestrains (50–300%) used in the assembly process, compressive forces (for example, $\sim 10\,$kPa) sufficiently large to reduce the out-of-plane dimensions of the interconnects to 30% of the original values (that is, a compression ratio of 0.3) lead to maximum principal strains in the metal layer that are below 1.3%, which is much smaller than the fracture strain ($> 5\%$) of gold or copper. This result indicates that the 3D helical interconnects can survive such compression ratios (0.3) and pressures $> 10\,$kPa. Experimental demonstrations of robust performance of 3D coil interconnect networks, involve simple test structures a LED to allow visual observation of operation. Each LED interconnect takes the form of a 3D helical coil (width 50 μm, thickness 1 μm), selectively bonded to an elastomeric substrate (Ecoflex 00–30, Smooth-on, USA; E = $\sim 30\,$kPa) using interface chemistries described previously and encapsulation with an ultra-soft

elastomer (Silbione 4717A/B, Bluestar Silicones, USA; $E = \sim 3$ kPa) as outlined above. The computational results and experimental results are in Supplementary Figs 28 and 29. The experiments show that an LED system constructed with networks of helical interconnects can survive different types of compressive loadings that might occur in practical use, consistent with modeling results.

Experimental measurements of the mechanical responses also show good agreement with modelling, even at the full system level of $\sim 250$ helical interconnects and $\sim 50$ chips (Fig. 3d–f). Overlays of optical images with FEA results (Supplementary Fig. 30) facilitate comparisons. Local regions can be examined quantitatively by using the index of structural similarity (SSIM)[54]. The results reveal SSIM indices of $\sim 0.81$, 0.75 and 0.77, respectively, for the cases of radial, $X$ axis and $Y$ axis stretching shown here (see Supplementary Fig. 31 for details). For reference, comparisons of images of local areas to themselves after translation or rotation (Supplementary Fig. 32) indicate that an SSIM value of 0.75 corresponds to a 2.3% relative $X$ axis offset, a 2.0% relative $Y$ axis offset or a 3.3° rotation relative to the image centre; an SSIM of 0.81 corresponds to a 1.8% $X$ axis offset, a 1.5% $Y$ axis offset or a 2.5° rotation.

### Demonstration of soft electronics built with 3D helical coil.
Figure 4a and Supplementary Figs 33–35 present optical images of the system in various states of deformation, with an FEA result for the rightmost frame. Functionally, devices with these designs offer reliable operation in all such circumstances (Supplementary Figs 34 and 35). Figure 4b,c shows capabilities in three-axis accelerometry for tracking of 3D motion and respiration. As in Fig. 4d, simultaneous monitoring of electrophysiological signals is also possible, including capture of electrocardiogram (ECG), electromyogram (EMG), electrooculogram (EOG) and electro-encephalogram (EEG) data for quantitative evaluation of cardiac, muscle, eye and brain activity, respectively. Multimodal operation depicted in Fig. 4e, Supplementary Fig. 36 and Supplementary Movie 1 involves recording of three-axis acceleration and EP signals simultaneously, where the former provides important contextual information on the latter. As shown by Supplementary Fig. 37, ECG data collected using a device in an undeformed state are identical to those collected using a device stretched radially to a strain of 50%. This invariance in operation is consistent with electrical resistances of the coils that remain constant under mechanical deformation. The 3D design approaches, the coupled mechanical/electrical considerations in layout and the two-stage encapsulation method, are each critically important in the properties and the operation of such systems.

### Discussion
The results presented here establish concepts, as well as routes for practical implementation, for 3D microstructure designs in soft electronics. Specific findings include quantitative advantages of ideal, spring-like mechanics in 3D helical coils compared to traditional 2D layouts; scalable approaches for forming 3D helical frameworks as interconnect networks between advanced microsystem components; combined electrical/mechanical techniques for optimizing system design; encapsulation materials and methods for ideal, 3D mechanics; and multimodal, wireless, skin-mounted demonstration devices for health monitoring. These ideas in 3D design have relevance not only to interconnect networks but also to other sub-systems, such as the antennas, the sensor structures and certain of the active and passive device components as well. Compatibility of the 3D assembly approaches with the most advanced methods in 2D micro/nanofabrication provides alignment both with state-of-the-art

microsystems technologies and unusual classes of materials and devices. A combination of established and developing elements in overall architectures that leverage both 2D and 3D layout features affords powerful opportunities not only in bio-integrated electronics but also in many other areas of emerging interest, from soft robotics to systems for virtual reality to hardware for autonomous navigation.

### Methods
**Finite element analysis.** Three-dimensional FEA techniques allowed prediction of the mechanical deformations and stress distributions of helical interconnects and entire circuit systems, during processes of compressive buckling and re-stretching. Four-node shell elements with a three-layer (PI/metal/PI) composite modelled the interconnects, and eight-node solid elements modelled the substrate and encapsulation. Refined meshes ensured computational accuracy using commercial software (Abaqus). The critical buckling strains and corresponding buckling modes determined from linear buckling analyses served as initial imperfections in the postbuckling analyses to determine the deformed configurations and strain distributions. To evaluate stretchability in the encapsulated condition, 3D helical interconnects determined by postbuckling analyses were embedded in an encapsulation solid that covered the entire area of substrate. The elastic stretchability corresponds to point at which the Mises stresses in the metal layer exceed the yield strength ($\sim 199$ MPa for Au and 357 MPa for Cu, both corresponding to $\sim 0.3\%$ yield strain) across at least one quarter of the width of any section of the interconnect. Previous experimental studies[32,55] support the use of this type of criterion. A typical hyper-elastic constitutive relation, that is, the Mooney–Rivlin law, captured the properties of the elastomeric substrate (elastic modulus $E_{substrate} = 20$ kPa and Poisson's ratio $v_{substrate} = 0.49$) and encapsulation material ($E_{encapsulation} = 3$ kPa and $v_{encapsulation} = 0.49$). The relevant material parameters are ($C_{10} = 2.68$ kPa, $C_{01} = 0.67$ kPa, $D_1 = 0.006$ kPa$^{-1}$) for the substrate and ($C_{10} = 0.40$ kPa, $C_{01} = 0.10$ kPa, $D_1 = 0.04$ kPa$^{-1}$) for the encapsulation. The other material parameters are $E_{Au} = 70$ GPa and $v_{Au} = 0.44$ for gold; $E_{Ni} = 200$ GPa and $v_{Ni} = 0.31$ for nickel; $E_{Cu} = 119$ GPa and $v_{Ni} = 0.34$ for copper; and $E_{PI} = 2.5$ GPa and $v_{PI} = 0.27$ for PI.

**Fabrication of networks of 3D helical coils for fundamental study.** Spin-casting poly(methyl methacrylate) (PMMA; $\sim 100$ nm in thickness, Microchem, USA) formed a thin sacrificial layer on a glass substrate. Spin-casting polyimide (PI; $1 \sim 3$ μm in thickness, Sigma-Aldrich, USA), depositing thin layers of metal by electron beam evaporation (Cr/Au or Cu, thickness 0.1–1 μm), performing photolithography, wet-etching, spin-casting another layer of polyimide followed by oxygen reactive ion etching defined a network of 2D serpentine structures, referred to here as the 2D precursor. Dissolving the PMMA by immersion in acetone for 10 min allowed retrieval of the 2D precursor onto the surface of a piece of water-soluble tape (Water-Soluble Wave Solder 5414, 3M, USA). Selective deposition of Ti (5 nm)/SiO$_2$(50 nm) by electron beam evaporation through a shadow mask (free-standing patterned sheet of a photodefinable epoxy with thickness of 0.1 mm; SU-8, Microchem, USA) defined sites for strong bonding to a silicone elastomer substrate formed by spin-casting and curing a prepolymer (thickness 1 mm; Ecoflex 00-50A/B, Smooth-on, USA) on a glass plate. A custom mechanical stage allowed application of precisely controlled levels of biaxial strain to this elastomer, selected using guidance from computation to achieve the required geometrical transformation from 2D to 3D. Laminating the 2D precursor onto a prestrained substrate and heating (10 min, 70 °C in an oven) the system activated formation of strong siloxane bonds between the patterned SiO$_2$ layer on the 2D precursor and the surface of the silicone substrate. Dissolving the tape by immersion in DI water and releasing the prestrain transformed the 2D precursors into an extended network of 3D helical coils via compressive buckling.

**Fabrication process for soft wireless electronics.** Processes similar to those described in the previous section, but with additional steps in spin-casting, metal deposition, photolithography and reactive ion etching yielded networks with electrically isolated crossing points and contact pads as interfaces to electronic components. A conductive alloy (In97Ag3, Indalloy 290, Indium Corporation, USA) enabled bonding of the contacts associated with the components to the pads in the interconnect network, while still in a 2D geometry on a bi-axially prestrained elastomer substrate. Partially releasing this prestrain followed by casting a uniform layer of an ultra-low-modulus elastomer (Silbione 4717A/B, Bluestar Silicones, France; $E = \sim 3$ kPa) encapsulated the entire system while preserving freedom of motion of the helical coils upon application of strain. This layer physically protected the system from the surroundings during handling and use.

**Circuit design for electrophysiological sensing module.** The electrocardiography (ECG) circuit used an instrumentation amplifier (INA333, Texas Instruments, common-mode rejection ratio = 100 dB) as a pre-amplifier for differential signal inputs to suppress common-mode noise. A driven ground with negative feedback allowed further noise reduction, thereby improving the signal-to-

noise ratio. A high-pass filter (4.8 Hz) and a low-pass filter (54.1 Hz) in a Sallen-Key topology, defined the passband of the circuit. A non-inverting amplifier magnified the filtered signal to provide an overall gain of 40 dB. The circuit used single-ended power at 3.3 V. A DC offset of 1.65 V allows capture of signal in the range between 0 and 3.3 V. An 8-bit analogue-to-digital converter integrated into the wireless chip (nRF51822, Nordic Semiconductor) enabled digital acquisition at a sampling rate of 250 Hz. The same circuit measured electrocardiogram (ECG) and electroencephalogram (EEG) data. The circuit also can perform electro-oculography (EOG) and electromyography (EMG) with a gain of 80 dB and a passband of 10–500 Hz and 0.5–20 Hz, respectively.

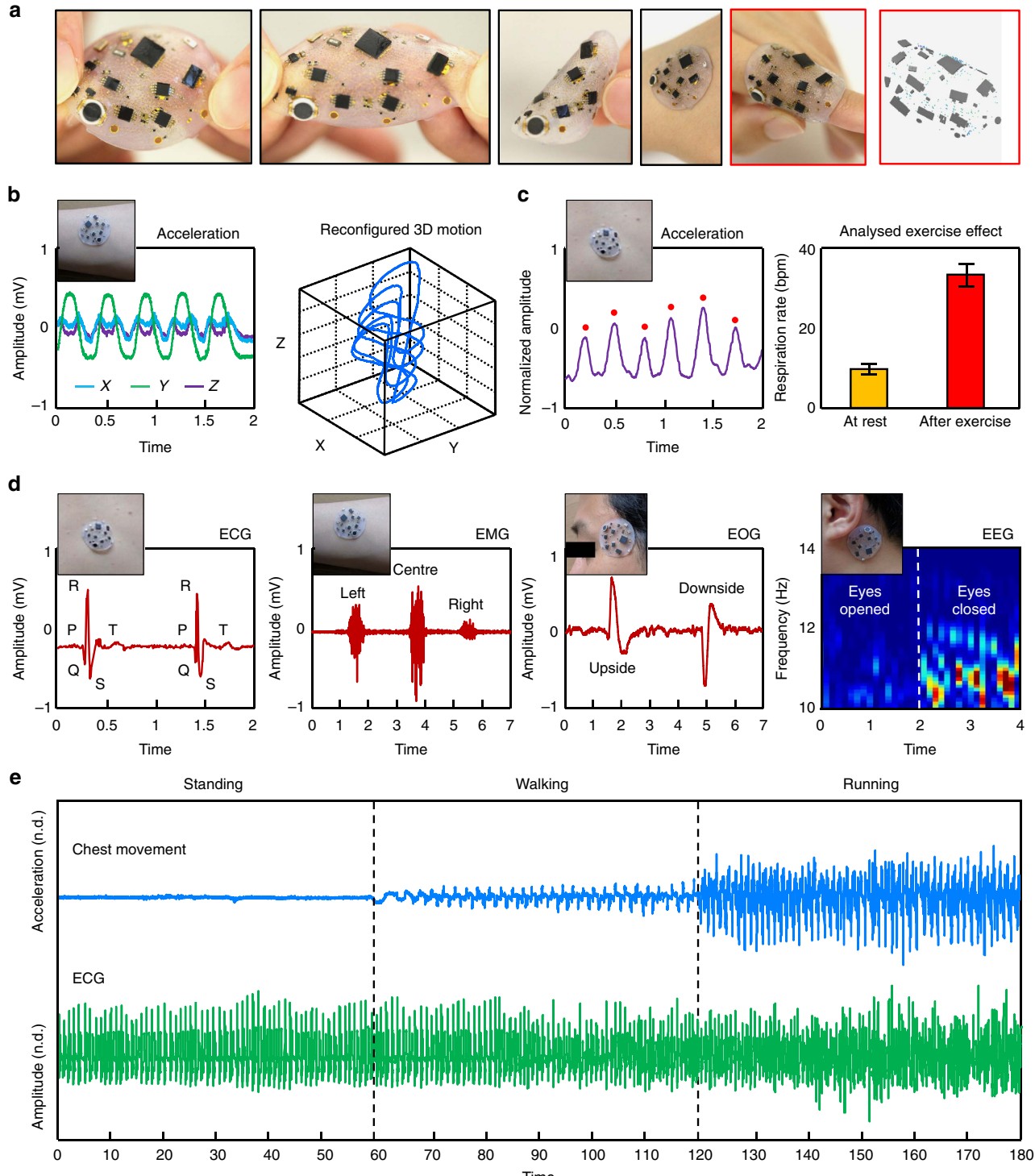

**Figure 4 | Mechanical and operational characteristics of the complete and encapsulated system.** (**a**) Optical images of a device deformed in different ways, with corresponding FEA results highlighted in red coloured boxes. (**b**) Representative recordings of three-axis acceleration from a device on the left forearm and inferred 3D patterns of motion. (**c**) Results for respiration rate extracted from frequency analysis of accelerometer data from a device placed on the chest, for cases at rest and after physical exercise. (**d**) Electrophysiological recordings with inset images of the device on the skin: electrocardiogram (ECG), electromyogram (EMG), electrooculogram (EOG) and electroencephalogram (EEG). (**e**) Wireless, multimodal monitoring of body activity, through simultaneous measurements of ECG (plotted in green colour) and movements of the chest by accelerometry (plotted in blue colour) during a time interval that includes standing (0–60 s), walking (60–120 s) and running (120–180 s).

**Circuit design for three-axis acceleration sensing module.** A digital accelerometer (ADXL345, Analog Devices) enabled measurements along three orthogonal axes. The configuration setup involved 13-bit resolution over a range of $\pm16\,g$, clock polarity = 1, and clock phase = 1 at a SPI clock speed of 2 MHz. The accelerometer operated at 3 V with bypass capacitors of 10 and 0.1 µF.

**Circuit design for wireless powering module.** Resonant inductive coupling using an unshielded wire wound inductor coil (27T103C, Murata Power Solutions Inc.) served as the basis for a wireless power receiver. A parallel capacitor (12–14 pF) defined a resonance at 13.56 MHz, as measured using an RF Impedance Analyzer (4291A, Hewlett Packard). A full-wave rectifier based on Schottky diodes and a smoothing capacitor rectified the received power. A low-dropout regulator (MIC5205, Microchip Technology) regulated the power at 3.3 V, to charge a supercapacitor (CPH3225A, Seiko Instruments) that operated the embedded system. The transmitter consisted of a circular wire wound coil with a radius of 3 mm, matched at 13.56 MHz, powered with an amplifier module (ID ISC.LR(M)2500, Feig Electronics) capable of delivering between 2 and 12 W. With this system, 60 mW can be transmitted at 2 W and a 1 cm distance, capable of operating the system, which had a peak power consumption of 30–40 mW (Supplementary Fig. 38).

**Programming the wireless embedded system.** Commercial packages including Keil µVision5 (ARM), Bluetooth Low Energy (BLE) at 2.4 GHz, S110 SoftDevice by Nordic Semiconductor served as tools for building software for the overall system. Wireless transmission of outputs from the ECG circuit and the accelerometer occurred in data packets with sizes of 6 bytes each. A custom Android-based application receives and processes signals from the transmitter. AChartEngine v1.1.0 (The4ViewSoftCompany) defined a graphical user interface with data logging function for further signal processing.

**Data availability.** The data that support the findings of this study are available from the corresponding authors on reasonable request.

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

## Acknowledgements

This work was supported by the Center for Bio-Integrated Electronics. K.-I.J. acknowledges the support from the Ministry of Science (#2017R1C1B2007795). Y.Z. acknowledges support from the National Natural Science Foundation of China (#11672152) and the National Basic Research Programme of China (#2015CB351900). Y.H. acknowledges the support from the NSF (#CMMI1300846, #CMMI1400169 and #CMMI1534120).

## Author contributions

K.-I.J., K.L., H.U.C., Y.H., Y.Z., J.A.R. designed idea and wrote manuscript. S.X., H.N.J., J.W.K., H.H.J., J.S., H.U.C., B.H.K., J.-H.K., J.L., K.J.Y., J.K., J.W.L., J.-W.J., Y.M.S. performed experiments and analysed the experimental data. K.L., Y.H. and Y.Z. led the structural designs and mechanics modelling, with assistance from C.Y., A.W. and Z.L. Others contributed to the analysis of experimental results.

## Additional information

**Competing interests:** The authors declare no competing financial interests.

