## [Peer Review File · Nature Communications]

Reviewers' comments:

Reviewer #1 (Remarks to the Author):

This work presents 3D architectures for soft electronics. Very interesting results are presented. This work deserves publication in Nature Communications after the following questions are addressed:

(1) After encapsulation, the stretch ability seems significantly reduced. What is the advantage of this method compared with others also developed by Rogers' group? Some quantitative comparisons are expected.

(2) It seems the out-of-plane deformation of the coils are huge and thus the rigidity in this direction is questionable. Even after encapsulation, the protection may not be sufficient. For example, will the device survive if someone put a finger and press the package? This test is expected.

(3) It seems that the coiled structures take many spaces. Can this one be designed denser and still have appreciable stretch ability?

Reviewer #2 (Remarks to the Author):

In this work, the authors present an integrated wearable sensing system that utilizes 3D interconnect mesh to enable high stretchability. The 3D interconnects is based on a novel buckling technology previously reported by the same group. This is the first work that applies this buckling concept for enabling a fully wearable sensing system consisting of integrated circuit (IC) chips. This is a clever work and presents a major advance in the field. I recommend that the paper be accepted for publication after minor revision notes below are addressed:

- please consider adding data on the electrical readout of the fully integrated system as the substrate is stretched. The authors may want to consider adding this data to the main figures.

- please consider adding scale bar to figure 3.

Reviewer #3 (Remarks to the Author):

Jang and co-authors described a novel device design and fabrication process for the 3D helical micro-coil interconnection. The 3D helical structure was fabricated by the compressive force induced on the 2D serpentine pattern in releasing its pre-strained planar form on an elastomeric substrate. The quantitative analysis using the finite element analyses (FEA) provided the optimum 3D configuration, which also proved the high stretchability of the non-coplanar 3D helical interconnection. In case of previous 2D serpentine structures, the mechanical stress was focused on the specific regions under their stretching deformations. On the other hand, in case of the 3D helical design, the stress was well distributed over the entire structure, which enables the highly compact and elastic soft electronic system. The encapsulation oftentimes limits stretchability. But a two-stage encapsulation strategy used in this work could overcome this issue. This process was optimized via computational modeling. The simulation certainly well matched with experimental results. Based on these experimental and theoretical efforts, the fully integrated circuit well achieved its original performance even under a variety of deformed states. One may argue that the 3D architecture through compressive buckling and fully integrated wearable sensor platform were similar with previous reports. However the complete system integration, in-depth understanding of underlying mechanics, and practical applications in physiological and electrophysiological monitoring provide a great deal of contributions in the field of epidermal soft electronics. Therefore, the reviewer recommends publication of this manuscript in Nature Communications. There are some minor comments that can be addressed before publication.

Comment #1: The authors have encapsulated the device by applying liquid precursor of the elastomer in prior to completely releasing the substrate's pre-strain. However, such an encapsulation process may affect the 3D pop-up procedure of ultrathin interconnections due to the viscosity of liquid. Any more detailed comments can be helpful.

Comment #2: In Figs. 2 and 3, demonstrations of the stable bonding to the bottom substrate under the compressive pop-up procedure and other mechanical deformations were shown. The 3D interconnection is so soft that its bonding can be well maintained. But the size of rigid chips is much bigger than interconnecting wires, and thereby there may be delamination of bonded chips under deformations. Further comments can be added on this issue.

Comment #3: It seems that Fig. 3a was not referred in the main manuscript.

Comment #4: Ecoflex was used for the bottom substrate and silibone was used for the top encapsulation. The reviewer is wondering any reason why two different materials are used, since the different mechanical and chemical properties may cause complexities.

Comment #5: In the Figure Caption, there is an error in Fig. 2d and e.

Responses to comments of Referee #1

Comments from Referee #1:

Summary Comment: “This work presents 3D architectures for soft electronics. Very interesting results are presented. This work deserves publication in Nature Communications after the following questions are addressed:”

Our response: We thank the referee for these positive comments and for the recommendation to publish with revisions. We carefully addressed the issues listed below, and we revised our manuscript accordingly.

Modification to the manuscript: None.

Comment 1: “After encapsulation, the stretch ability seems significantly reduced. What is the advantage of this method compared with others also developed by Rogers' group? Some quantitative comparisons are expected.”

Our response: We thank the referee for this comment. The encapsulation serves to protect the devices from direct exposure to the environment, and specifically from mechanical damage that could result from physical contact. As compared with liquid encapsulation (Science 2014, Vol 344: 70), the solid encapsulation adopted in this paper avoids issues in leakage or evaporation that can occur during long-time usage with liquid based systems. Serpentine and fractal-inspired designs are two representative interconnect technologies developed by our group. Quantitative mechanics modeling (Supplementary Figs 4-6) indicate that under identical encapsulation conditions, 3D helical interconnects outperform 2D serpentine designs that undergo buckling deformations in terms of local wrinkling, typically by a factor > 2.3 for representative prestrains ($> 100\%$). Even when compared to thick serpentine interconnects that are dominated by planar, scissor-like mechanics, 3D helical interconnects with sufficient prestrain (e.g., $> 100\%$) provide a considerable enhancement (e.g., > 1.8 times) in elastic stretchability. Such improvements, at an even greater factor, are also present in comparison to 2D interconnects with fractal-inspired layouts (Supplementary Fig. 7). We added text to clarify these points.

Modification to the manuscript: On Page 8, we added “In particular, 3D helical interconnects outperform 2D serpentine designs that undergo buckling deformations in the form of local wrinkling, typically by a factor > 2.3 for representative prestrains ($> 100\%$). As compared to thick serpentine interconnects that are dominated by planar, scissor-like mechanics, 3D helical interconnects with a sufficient prestrain (e.g., $> 100\%$) exhibit a considerable enhancement (e.g., > 1.8 times) in the elastic stretchability. Similar improvements, at an even greater factor, apply relative to 2D interconnects with fractal-inspired layouts (Supplementary Fig. 7).”

On Page 11, we added “As compared with liquid encapsulation¹⁴, the solid encapsulation approach adopted in this paper avoids the potential for leakage or evaporation.”

Comment 2: “It seems the out-of-plane deformation of the coils are huge and thus the rigidity in this direction is questionable. Even after encapsulation, the protection may not be sufficient. For example, will the device survive if someone put a finger and press the package? This test is expected.”

Our response: We thank the referee for this comment. We agree that the devices might encounter out-of-plane compressive forces due to physical contact during practical use. We performed additional experiments and simulations to investigate behaviors in such cases. Computations show that for helical interconnects assembled with a variety of prestrains (50% to 300%), compressive forces (e.g. ~10 kPa) sufficiently large to reduce the out-of-plane dimensions of the interconnects to 30% of the original values (i.e. a compression ratio of 0.3) lead to maximum principal strains in the metal layer that are below 1.3%, which is much smaller than the fracture strain (> 5%) of gold or copper.

Figure Caption: The pressure needed to compress the 3D helical interconnects such that the out-of-plane dimensions reach 30% of the original values as a function of prestrain used to form the coils.

A new set of experiments was carried out to demonstrate that a LED system constructed with networks of helical interconnects can survive different types of compressive loadings. To demonstrate robust performance of the electrical system consisting of 3D coil network, a LED system was prepared. Each LED chip is connected with 3D helix coils (width 50 μ m, thickness 1 μ m), which are selectively bonded on an elastomeric substrate (Ecoflex 00-30, Smooth-on, USA; $E \sim 30$ kPa) with siloxane bonding. The entire architecture is fully encapsulated with ultra-soft elastomer (Silbione 4717 A/B, Bluestar Silicones, USA; $E \sim 3$ kPa) to allow relatively free deformations of 3D microstructures and protect them from mechanical condition or impact such as compressive loading. The results are illustrated in the figure below.

Figure Caption: A stretchable test structure consisting of a pair of LEDs electrically connected by a 3D helical coil encapsulated in and supported by an elastomer (a). Compressive loads applied with the tip of a pen (b), the end of a pair of tweezers (c) and an index finger (d). The I-V curves under different conditions are in (e).

Modification to the manuscript: On Pages 13 and 14, we added “The devices might encounter compression along the thickness direction during practical use, associated with physical contact or normal impact. Quantitative modeling of the encapsulated helical interconnects provides insights into the mechanics associated with such situations. For a variety of prestrains (50% to 300%) used in the assembly process, compressive forces (e.g. ~ 10 kPa) sufficiently large to reduce the out-of-plane dimensions of the interconnects to 30% of the original values (i.e. a compression ratio of 0.3) lead to maximum principal strains in the metal layer that are below 1.3%, which is much smaller than the fracture strain ($> 5\%$) of gold or copper. This indicates that the 3D helical interconnects can well survive this level of compression ratio (0.3) and pressure larger than 10 kPa. To demonstrate robust performance of the electrical system consisting of 3D coil network, a LED system was prepared. Each LED chip is connected with 3D helix coils (width 50 μ m, thickness 1 μ m), which are selectively bonded on an elastomeric substrate (Ecoflex 00-30, Smooth-on, USA; $E \sim 30$ kPa) with siloxane bonding. And the entire architecture is fully encapsulated with ultra-soft elastomer (Silbione 4717 A/B, Bluestar Silicones, USA; $E \sim 3$ kPa) to allow relatively free deformations of 3D microstructures and protect them from mechanical condition or impact such as compressive loading. The computational results and experimental demonstration are summarized in Supplementary Figs 28 and 29. The experiment showed that a LED system constructed with networks of helical interconnects can survive different types of compressive loadings.”

We added the above figures as Supplementary Figs 28 and 29 in the revised manuscript.

Comment 3: “It seems that the coiled structures take many spaces. Can this one be designed denser and still have appreciable stretch ability?”

Our response: We thank the referee for this useful comment. Yes, the circuit can be designed to greater levels of density and still offer appreciable levels of stretchability. One method is to adopt an increased prestrain during the 3D assembly process. By increasing the prestrain from 150% to 200% for example, the area of entire circuit can be reduced to $\sim 70\%$ of the original area. The elastic stretchability undergoes relatively

small reductions for this case, as shown below. We added relevant text and figures to clarify this point.

Figure Caption: Design of circuit system to increase the areal density. (a) FEA result for the deformed configuration of a system with prestrain ϵ_{pre} (200%). (b) Uniaxial and radial elastic stretchability for two prestrains (150% and 200%), and the same encapsulation strain (30%). The color in (a) denotes the Mises stress.

Modification to the manuscript: On Page 13, we added “The circuit can be designed with improved density via the use of increased prestrain during the 3D assembly process. By increasing the prestrain from 150% to 200%, the area of entire circuit can be reduced to $\sim 70\%$ of the original area. The elastic stretchability undergoes a relatively small corresponding reduction, as in Supplementary Fig. 27.”

We added the above figure as Supplementary Fig. 27 in the revised manuscript.

Responses to comments of Referee #2

Comments from Referee #2:

Summary Comment: “In this work, the authors present an integrated wearable sensing system that utilizes 3D interconnect mesh to enable high stretchability. The 3D interconnects is based on a novel buckling technology previously reported by the same group. This is the first work that applies this buckling concept for enabling a fully wearable sensing system consisting of integrated circuit (IC) chips. This is a clever work and presents a major advance in the field. I recommend that the paper be accepted for publication after minor revision notes below are addressed.”

Our response: We thank the referee for these positive comments and the recommendation to publish with minor revisions.

Modification to the manuscript: None.

Comment 1: “please consider adding data on the electrical readout of the fully integrated system as the substrate is stretched. The authors may want to consider adding this data to the main figures.”

Our response: We thank the referee for this useful comment. As recommended, we collected ECG data from a device fully stretched by use of a customized biaxial stretcher. The filtered ECG signals do not depend on the extent of stretching because the electrical resistance values of coils in the device do not change under induced mechanical deformation. We placed these data in figure S32 and we added associated text to the manuscript.

Figure Caption: Collected ECG signal from the chest using a device in an (a) undeformed configuration and (b) in a state of radial stretching to a strain of 50%.

Modification to the manuscript: Page 15, we added “As shown by Supplementary Fig. 37, ECG data collected using device in an undeformed state are identical to those collected using a device stretched radially to a strain of 50%. This invariance is consistent with electrical resistances of the coils that remain constant under mechanical deformation.”

We added the above figure as Supplementary Fig. 37 in the revised manuscript.

Comment 2: “please consider adding scale bar to figure 3.”

Our response: We added scale bars in figure 3 and its caption.

Modification to the manuscript: (Figure 3 caption) Scale bars of (a), (b), and (d) are 1 cm. The scale bar of the inset of (a) is 1 mm.

Responses to comments of Referee #3

Comments from Referee #3:

Summary Comment: “Jang and co-authors described a novel device design and fabrication process for the 3D helical micro-coil interconnection. The 3D helical structure was fabricated by the compressive force induced on the 2D serpentine pattern in releasing its pre-strained planar form on an elastomeric substrate. The quantitative analysis using the finite element analyses (FEA) provided the optimum 3D configuration, which also proved the high stretchability of the non-coplanar 3D helical interconnection. In case of previous 2D serpentine structures, the mechanical stress was focused on the specific regions under their stretching deformations. On the other hand, in case of the 3D helical design, the stress was well distributed over the entire structure, which enables the highly compact and elastic soft electronic system. The encapsulation oftentimes limits stretchability. But a two-stage encapsulation strategy used in this work could overcome this issue. This process was optimized via computational modeling. The simulation certainly well matched with experimental results. Based on these experimental and theoretical efforts, the fully integrated circuit well achieved its original performance even under a variety of deformed states. One may argue that the 3D architecture through compressive buckling and fully integrated wearable sensor platform were similar with previous reports. However the complete system integration, in-depth understanding of underlying mechanics, and practical applications in physiological and electrophysiological monitoring provide a great deal of contributions in the field of epidermal soft electronics. Therefore, the reviewer recommends publication of this manuscript in Nature Communications. There are some minor comments that can be addressed before publication.”

Our response: We thank the referee for these positive comments and the recommendation to publish with revisions. We carefully addressed the issues detailed below, and revised our manuscript substantially.

Modification to the manuscript: None.

Comment 1: “The authors have encapsulated the device by applying liquid precursor of the elastomer in prior to completely releasing the substrate’s pre-strain. However, such an encapsulation process may affect the 3D pop-up procedure of ultrathin interconnections due to the viscosity of liquid. Any more detailed comments can be helpful.”

Our response: We thank the referee for this comment. As can be seen in the picture below, the 3D buckled structures in the system maintain their original form because the equivalent moduli of the buckled materials are sufficiently high to remain largely undeformed by shear forces associated with the viscosity of the uncured elastomer or liquid.

Figure Caption: Optical images of 3D coil after encapsulation with ultra-soft elastomeric materials (Silbione 4717 A/B, Bluestar Silicones). The 3D structure keeps their original structures well. Scale bars of (a) and (b) are 1mm and 100um, respectively.

Modification to the manuscript: On Page 12, we added “As illustrated by experimental results (Supplementary Fig. 24), the 3D buckled structures in the system maintain their original forms during the encapsulation process because their equivalent moduli are sufficiently high that they are not deformed by shear forces associated with the viscosity of the uncured elastomer or liquid.”

We added the above figure as Supplementary Fig. 24 in the revised manuscript.

Comment 2: “In Figs. 2 and 3, demonstrations of the stable bonding to the bottom substrate under the compressive pop-up procedure and other mechanical deformations were shown. The 3D interconnection is so soft that its bonding can be well maintained. But the size of rigid chips is much bigger than interconnecting wires, and thereby there may be delamination of bonded chips under deformations. Further comments can be added on this issue.”

Our response: We thank the referee for this comment. We agree that the delamination might occur at the chip/substrate interface during the process of mechanical-guided assembly. In general, the driving forces for delamination depend mainly on the magnitude of the prestrain, the sizes of the chips and their effective moduli, for a given substrate. The delamination can occur more easily as the prestrain increases or the effective rigidity of the chip increases. Such trends are in qualitative agreement with experimental observations (as shown below), in which the delamination area increases with the effective rigidity of the chip increases. In addition, delamination occurs mainly at the periphery regions of the chips. This type of partial delamination does not affect the bonding conditions between the electrical interconnects and the substrate, and thereby has a minor effect on the integrity of the entire device. We added text to clarify this issue.

Figure Caption: Potential delamination at the chip/substrate interface after release of prestrain: (a) fully laminated, (b) partially delaminated, to a slight degree, (c) partially delaminated to a large degree. Even for the case of slight delamination, the electrical interconnection remains stable.

Modification to the manuscript: On Pages 10 and 11, we added “Delamination has the potential to occur at the chip/substrate interface during the process of mechanical-guided assembly, as shown in experimental results (Supplementary Fig. 19). Theoretically, delamination occurs more readily as the prestrain increases or the effective rigidity of the chip increases. Such trends are in qualitative agreement with experimental observations (Supplementary Fig. 19), in which the delamination area increases with the effective rigidity of the chip increases. In addition, delamination occurs mainly at the periphery regions of the chips. This type of partial delamination does not affect the bonding conditions between electrical interconnects and substrate, and thereby has a minor effect on the integrity of the entire device.”

We added the above figure as Supplementary Fig. 19 in the revised manuscript.

Comment 3: “It seems that Fig. 3a was not referred in the main manuscript.”

Our response: We added text to refer to Fig. 3a in the revised main manuscript.

Modification to the manuscript: On Pages 11 and 12, we changed to “An alternative, improved strategy that naturally follows from the 3D assembly process outlined in Figure 1 involves applying this material as a liquid precursor at a state of partial release of the substrate prestrain, ϵ_{encap} , crosslinking it into a solid form and then completing the release, as shown in Fig. 3a.”

Comment 4: “Ecoflex was used for the bottom substrate and silibone was used for the top encapsulation. The reviewer is wondering any reason why two different materials are used, since the different mechanical and chemical properties may cause complexities.”

Our response: We thank the referee for this comment. Quantitative mechanics simulations indicate that the modulus of top encapsulation has a significant effect on the elastic stretchability of 3D helical interconnects, as shown by the figure below. We selected the low modulus material Silbione for the top encapsulation, mainly to minimize mechanical constraints on the deformations of helical interconnects, thereby achieving a stretchability much higher than that provided by alternative materials with higher modulus, such as Ecoflex. We added text and new results to clarify this point.

Figure Caption: Effect of the modulus of the encapsulation material on the elastic stretchability of 3D helical interconnects. Elastic stretchability versus the prestrain for materials with a range of modulus values (from 0 to 20 kPa). The parameters adopted in the simulations are ($\theta_0=180^\circ$, $w=50 \mu\text{m}$, $t_{\text{metal}}=0.6 \mu\text{m}$, $t_{\text{PI}}=6.0 \mu\text{m}$, $E_{\text{substrate}}=20 \text{ kPa}$, $E_{\text{metal}}=200 \text{ GPa}$).

Modification to the manuscript: On Page 13, we added “The modulus of the top encapsulation has a significant effect on the elastic stretchability of the 3D helical interconnects (Supplementary Fig. 26). We selected a low modulus elastomer (Silbione) to reduce mechanical constraints on the deformations of helical interconnects, thereby achieving a much higher stretchability than possible with elastomers with higher modulus values (e.g. Ecoflex).”

We added the above figure as Supplementary Fig. 26 in the revised manuscript.

Comment 5: “In the Figure Caption, there is an error in Fig. 2d and e.”

Our response: We corrected the relevant caption in the revised manuscript.

Modification to the manuscript: Caption of Figure 2, we changed to “(e) Optical image of a device supported by two fingers.”

REVIEWERS' COMMENTS:

Reviewer #1 (Remarks to the Author):

The revised manuscript has addressed my previous comments. Publication as it was suggested.

Reviewer #2 (Remarks to the Author):

The authors have sufficiently addressed all concerns. I recommend its publication in its present form.

Reviewer #3 (Remarks to the Author):

All comments from the reviewer were thoroughly addressed, and the revised manuscript is now ready for publication.

Reviewers' comments:

Reviewer #1 (Remarks to the Author):

The revised manuscript has addressed my previous comments. Publication as it was suggested.

Reviewer #2 (Remarks to the Author):

The authors have sufficiently addressed all concerns. I recommend its publication in its present form.

Reviewer #3 (Remarks to the Author):

All comments from the reviewer were thoroughly addressed, and the revised manuscript is now ready for publication.

Our response: We thank the all referees for these positive comments and recommendation to publish our manuscript in *Nature Communications*.